# Mapping the Protein Kinome: Current Strategy and Future Direction

**DOI:** 10.3390/cells12060925

**Published:** 2023-03-17

**Authors:** Zhanwu Hou, Huadong Liu

**Affiliations:** 1Center for Mitochondrial Biology and Medicine, Douglas C. Wallace Institute for Mitochondrial and Epigenetic Information Sciences, The Key Laboratory of Biomedical Information Engineering of Ministry of Education, School of Life Science and Technology, Xi’an Jiaotong University, Xi’an 710049, China; 2School of Health and Life Science, University of Health and Rehabilitation Sciences, Qingdao 266071, China

**Keywords:** kinome, phosphorylation, activity assay, spatial, proximity labeling, low cell numbers

## Abstract

The kinome includes over 500 different protein kinases, which form an integrated kinase network that regulates cellular phosphorylation signals. The kinome plays a central role in almost every cellular process and has strong linkages with many diseases. Thus, the evaluation of the cellular kinome in the physiological environment is essential to understand biological processes, disease development, and to target therapy. Currently, a number of strategies for kinome analysis have been developed, which are based on monitoring the phosphorylation of kinases or substrates. They have enabled researchers to tackle increasingly complex biological problems and pathological processes, and have promoted the development of kinase inhibitors. Additionally, with the increasing interest in how kinases participate in biological processes at spatial scales, it has become urgent to develop tools to estimate spatial kinome activity. With multidisciplinary efforts, a growing number of novel approaches have the potential to be applied to spatial kinome analysis. In this paper, we review the widely used methods used for kinome analysis and the challenges encountered in their applications. Meanwhile, potential approaches that may be of benefit to spatial kinome study are explored.

## 1. Introduction

Protein kinases (PKs) are key regulators of cell function, and they are also one of the largest and most functionally diverse gene families [1]. Humans contain around 560 PKs, of which about 500 eukaryotic PKs are divided into eight major groups and about 60 atypical PKs [2]. PKs catalyze the transfer of the γ phosphate group from ATP to a protein acceptor, mainly to serine, threonine, and tyrosine residues. Then, the negatively or positively charged amino acids, which are close to the phosphorylated residues, will be repulsed or attracted, respectively. The phosphate group will introduce a very hydrophilic and polar region in the affected protein. It may lead to conformational changes in the proteins, altering their functions or interactions with other biomolecules [3]. The phosphorylation network mediated by PKs is involved in nearly all eukaryotic cellular pathways. Thus, PKs are crucial to cell growth and development [4]. Dysregulation of kinase signaling, therefore, leads to cancers and a variety of other human disorders, including neurological, metabolic, cardiovascular, infectious, and immunological diseases [5,6]. As a result, PKs have been recognized as the most attractive pharmaceutical targets. Indeed, an increasing number of kinase inhibitors have been developed and used for cancer therapy [7].

There are currently 72 small molecule kinase inhibitors [8] approved by the FDA for the treatment of disease (Figure 1). Unfortunately, fewer than 10% of all PKs are targeted by FDA-approved inhibitors. Some challenges limit the full potential of kinases as drug targets, including validating novel kinase targets [9], obtaining target selectivity to reduce off-target-mediated toxicity [10], and developing efficient compound screening technologies [11]. Overcoming these limitations relies on sensitive kinase activity detection to discover new targets, to profile inhibitor selectivity, and to perform inhibitor screening [12]. With the efforts of academia and industry, sensitivity and high-throughput detection of most kinase activities have been achieved by monitoring the phosphorylation of substrate peptides with a variety of sensitive detection methods, such as fluorescence, electrochemistry, and absorbance [13,14]. The existence of an integrated kinase network has spurred the focus onto the kinome rather than individual kinases [15]. A more complete kinome profile and better knowledge of the events regulated by the kinome will facilitate the design and application of targeted kinase inhibitors.

To map the kinome profile, massive technologies for assessing kinome activities have been established. In particular, the technique that measures the kinome profile in a physiological environment is an important milestone. This review aims to collect and provide insight into these methods and highlight the opportunities and challenges posed by each method. Additionally, with the research on spatial-temporal omics, we introduce some strategies that may expand rapidly into this area and enable further kinome studies.

## 2. Technologies for Kinome Analysis

The kinome profile correlates with the inhibitor design and therapy strategy. Therefore, numerous techniques have been developed to compare kinome activities in normal and aberrant tissues. Such techniques include a kinome peptide library based on kinase substrates, kinome enrichment that relies on kinase inhibitor conjugated beads, a reactive chemical probe for kinome competition labelling, and kinome activity-representing phosphorylation sites. Each method has its own applications and limitations, but it is certain that a combination of these techniques can cover a larger portion of the kinome than any technique alone.

### 2.1. Kinome Substrate Peptide Library

Monitoring kinome substrate library phosphorylation sensitively is quite an effective way to represent kinome activity [16]. As the physiological substrates of most kinases are proteins with considerable diversity, this poses a serious challenge in constructing a stable kinase substrate library. Most kinases recognize and modify their targets based on residues near the phosphorylation site through highly conserved catalytic mechanisms, which is largely in the disordered region. In particular, the maximum reaction rate (Vmax) and Michaelis–Menten constant (Km) of the catalytic reaction with peptide mimic kinase substrates were close to those of native protein substrates [17]. This makes it possible to monitor kinase activity using short peptides of specific sequence as alternative substrates. The use of peptides for kinome analysis has great advantages such as simple synthesis, low cost, and high chemical stability. Based on these advantages, the use of a kinome substrate peptide library (KsPL) to quantify kinome activity has been promoted [18,19].

There are a variety of kinome detection methods based on KsPL. They can be classified into three categories: planar peptide array, 3D peptide array, and an in-solution peptide library (Figure 2) [16]. The planar peptide array refers to kinome detection, which is formed by attaching peptides to a planar physical support. To evaluate the reproducibility of the technique, each peptide is printed as multiple spots. The 3D peptide array is the peptide library immobilized on a 3D support [20]. Compared to the conventional planar array, the 3D support provides a larger surface area for peptide presentation and a higher reaction rate, improving the sensitivity and accuracy of the kinome activity assessment [21,22]. The activated alumina surface used as a support for the 3D array is called PamChip [23]. This technique has been applied for a number of important studies including kinome profiling of endothelial inflammation [24], mechanisms of drug resistance [25] and druggable targets identification for cholangiocarcinoma [26]. The disadvantage is that it displays fewer spots compared to a planar array, which has more than 1000 spots. An in-solution peptide library is another option for assessing kinome activity. The two methods mentioned above for immobilizing the peptides may prevent some of the normal enzyme–substrate interactions. An in-solution library enables peptides to achieve the kinome easily and react with the kinome completely, and the application of LC-MS/MS greatly improves its sensitivity [27]. However, the cumbersome steps and expensive instruments limit its wide application. Although differences exist in each method of peptide library display, they all evaluate kinome activity based on the phosphorylation of substrate peptides. Typically, the peptide library is incubated with cell lysate, and active kinases in cell lysates phosphorylate the corresponding substrates in the peptide library. The phosphorylated peptides can be visualized and quantified by radioautography, Western blotting with phosphorylation-specific antibodies, fluorescence spectroscopy, and mass spectrometry (MS) in high-throughput manor [28,29]. Finally, the relative kinome activity in the given lysate can be evaluated [30].

Designing a proper peptide library for kinome analysis is one of the most crucial aspects in ensuring its accuracy. The proper peptide can be screened via one-bead-one-compound [31], phage display [32], or PeSA [33]. The PeSA is a software tool, which uses the peptide array data for sequence analysis. It has the potential for substrate peptide generation. These design methods can generate suitable peptides without the limitation of known sequences. However, they are time-consuming and labor-intensive and this limits their widespread application. The peptide library information can also be obtained from previous phosphoproteome investigations, or from publicly available databases such as PhosphoSitePlus [34] or PhosphoELM [35]. However, online databases contain many phosphorylation sites for the human and mouse, but have only limited information available for other species. Besides, phosphorylation sites are not fully conserved between human and other species. Thus, many computational tools have been developed to enable the prediction of phosphorylation events based on phosphorylation events described in other species with a similar sequence [36]. To date, species-specific peptide libraries have been created for cattle, pigs, rodents, chickens, horses and dogs through the application of computational platforms [37]. DAPPLE and KinasePhos are software pipelines that allow users to identify potential phosphorylation sites easily, quickly and accurately in an organism of interest [38,39].

Most of the advanced technologies in kinome analysis are related to data analysis methodologies. As the data show similarity, the kinome data generated through a peptide array are processed by a software platform, which is designed to analyze the microarray gene expression data. However, the quantity of data from gene expression and kinome analyses varies greatly, which makes the statistical criteria used to evaluate gene expression inapplicable to kinome studies [40]. To alleviate these limitations, some software platforms are specifically designed for analyzing peptide array data. PIIKA is capable of identifying known pathways with much greater statistical confidence [41]. Kinomics Toolkit provides users with a platform for exploring peptide array data [42]. KRSA has been utilized to analyze the kinome signature of mice with a genetic deletion [43]. These established new strategies and software platforms for KsPL particularly promote the development of kinome analysis.

To date, KsPL is used to study a variety of biological events. The species-specific kinome peptide arrays were designed for bovine species to study Salmonella within the peripheral lymph nodes [44]. The chicken-specific peptide arrays were used to analyze the kinome alteration of host responses to Salmonella [45]. These studies have contributed to the development of cost-effective, pre-harvest intervention strategies to prevent pathogen infection. A system kinome approach has been used to compare host cell responses to different monkeypox viruses, which differ in terms of severe infection and minimal lethality. The results suggested that Akt and p53 might be potential host therapeutic targets [46]. Array-based kinome profiling revealed that signals were rewired in resistant cells, and the results were confirmed by Western blots [47]. Furthermore, signaling pathways with spatial resolution have been achieved by analyzing the kinome in subcellular partitioning [48].

### 2.2. Kinase Inhibitor Conjugated Beads

Using kinase inhibitor conjugated beads to enrich PKs from total cell extracts is an important way to improve the sensitivity of the kinome analysis. It has been used for the simultaneous enrichment of the kinome (Figure 3) [49]. By immobilizing multiple pan-kinase inhibitors with different specificities and affinities on agarose beads, kinase inhibitor conjugated beads enabled the capture of a large proportion of the kinome in a single experiment. Then, the captured kinome was subsequently eluted and identified by MS or Western blotting [50]. Many kinase inhibitors have been derivatized to facilitate rapid coupling to modified agarose or other supports without losing binding groups. In addition, inhibitor bead compositions have been customized to meet the needs of specific experiments [51].

The selection of inhibitors or inhibitor derivatives is essential to improve the depth of the kinome assay. Kinase capture with CDK or p38 inhibitors has been used to identify CDK or p38-associated kinases. Some novel intracellular targets can also been identified by that method [52,53]. Unfortunately, only 30 kinases have been identified. An improvement in the original kinase capture has resulted in a multichannel inhibitor beads (MIBs) approach, which uses individual, layered, immobilized kinase inhibitors in a column [54]. The use of a layer-wise approach allows more abundant kinases to be bound by highly selective inhibitors, providing additional binding “space” for lower abundance kinases. Thus, the kinome detection depth can be further improved. The original components of MIBs from top to bottom are Bisindoylmaleimide-X, SB203580, Lapatinib, Dasatinib, Purvalanol B, VI16832, and PP58 [55]. This approach has revealed 50–60% of expressed kinome activity, and identified a reprogramming of the kinome in response to MEK inhibitors. More than 220 kinases are obtained from 60 cell lines using MIBs, which are then verified by MS-based protein identification [56]. Although kinase enrichment with broad-spectrum kinase inhibitors has enabled the identification of a large number of kinases, it also has shortcomings. Especially, the available inhibitors are insufficient to cover the entire kinome [57]. To address this limitation, researchers have used complementary chemical probes to optimize kinome coverage [58]. For example, in a mixture of seven kinase inhibitors that lack a suitable affinity for Akt kinase, the Akt selective inhibitor GSK69069327 is used to achieve Akt kinase coverage [59]. Some specific inhibitors are also used to study related kinases, such as VEGFR [60] and FGFR [61], and novel inhibitors are being developed to extend the kinome coverage.

Several quantification methods for an enriched kinome have been explored and which contributed to a wide application of kinase inhibitor conjugated beads. Data dependent acquisition (DDA) is a typical mode used for kinome quantification, and around 110–125 kinases can be assayed in any single LC-MS/MS run [62]. In addition, isobaric tags, such as TMT and iTRAQ, depending on DDA mode, have been used for the relative quantitation of downstream MIBs enrichment [63]. SILAC labeling based on DDA is also an alternative means of quantification and has been used in the study of kinases associated with the cell cycle [64]. However, despite kinome enrichment being performed, quantification based on DDA mode still suffers from highly complex peptide mixtures. In addition, it is not conducive to the quantitative detection of low abundance kinases. Data independent acquisition (DIA) via Sequential Window Acquisition of all Theoretical fragment ions (SWATH) can improve the detection of low-abundance proteins. SWATH increases the number of kinase identifications relative to DDA [65]. DIA via parallel reaction monitoring (PRM), which is a targeted proteomic methodology, allows for the quantitation of a selected set of target peptides with an improved sensitivity in detection. A PRM strategy, which exploits meter-scale monolithic silicone-C18 column chromatography, has demonstrated the reliable quantification of more than 150 kinases in a single run [66]. The kinome detection coverage of each method is often not identical, so the combination of multiple methods can often further increase kinome detection [67].

The advantage of this method is its ability to capture and quantify hundreds of kinases in a single experiment, especially in members of the kinome that have not been studied. Lots of studies have shown that the inhibitor beads enrich the kinome according to kinase activity. However, more caution should be exercised when inferring kinase activity from such binding data. The experiments of quantitative phosphoproteomics and the purification of the kinome from signaling activated or resting cancer cells indicate that this is not the case [68]. The kinase’s activity will affect its affinity to the beads, but the abundance of the kinase and the type of inhibitor have a greater effect, which benefits the further development of this method.

As is known, kinase inhibitor conjugated beads have been widely utilized to study kinome profiling in cancer. Some driver kinases and putative therapeutic targets were revealed by the kinome landscape of high-grade serous ovarian carcinoma, and several kinases were previously unexplored, such as MRCKA [69]. The application of inhibitor conjugated beads revealed that kinome reprogramming limited the efficiency of the MEK inhibitor, and inhibition of BET proteins could block reprogramming and enhance the therapeutic efficacy [70]. Wee1 is a survival factor in the treatment of gastrointestinal stromal tumors (GIST) with avapritinib, and the combination of MK-1775 (Wee1 inhibitor) and avapritinib have a notable efficacy in inhibiting GIST cell lines [71].

### 2.3. Chemical Reactive Probe

The chemical probe labeling strategy is a widely used method in kinome analysis [72]. Such chemical probes used in the kinome assay generally have three main functions, which are the selectivity function—to recognize the functional domain of the target protein; reactivity function—forms irreversible covalent binding with the target protein after activation; sorting function—enables the capture of the kinome from complex protein mixtures (Figure 4a) [73]. This labeling method has a wide application in the purification, enrichment and identification of the kinome [74]. The most commonly used probe for kinome analysis is the desthiobiotin-ATP affinity probe (Figure 4b) [75]. The design of this probe is based on two important characteristics of kinase. One is that most kinases contain an ATP-binding domain that recognizes ATP specifically and broadly. The other is that most kinases have at least one conserved lysine residue within the active site indicated by sequence comparisons [76]. Therefore, the desthiobiotin-ATP recognizes the ATP-binding domain of the kinase and binds to the pocket. The acyl phosphate reactive group of the probe will be close and react with the lysine residue to yield a stable amide bond, which results in the covalent attachment of desthiobiotin together with a linker to the kinase. Finally, the bound peptides or proteins in the composite samples are enriched by streptavidin resin and identified by MS [77]. The probe was first reported in 2006 and the ATP analogue FSBA was used for target recognition. A total of 132 proteins were identified by the probe, of which 6 were protein kinases [78]. Subsequently, ATP was used in the probe to expand kinome identification, and nearly 400 PKs were identified in various tissues and cell lines, covering 80% of the kinome [79]. An AMP-Biotin probe is used to explore novel AMP-binding kinase, which can expand the kinome coverage [80].

Besides ATP, there are other small molecules that can also be used as recognition elements for the design of kinome probes. Kinome probe based on Staurosporine, which is an ATP analog, can capture more than 100 kinases in HepG2 cells [73]. Dasatinib is a dual Bcr-Abl and Src family tyrosine kinase inhibitor, which has been approved by the FDA for the treatment of tumors. A dasatinib-based probe has enabled a proteome-wide profiling of potential dasatinib cellular targets, among which were several previously unknown serine/threonine kinases [81]. The intracellular kinome can be covalently labeled with a sulfonyl fluoride probe, which is named XO-44. It can recognize the ATP-binding domain and react with the conserved lysine. After capturing the labeled proteins, up to 133 endogenous kinases have been identified [82]. KY-26, which has been modified based on XO-44, performs better in kinome identification [83]. Although probes have been developed to cover most kinases, it is still necessary to develop specific probes for hard-to-detect kinases. These specific probes can be designed according to the structure of the kinase. The amino acid used in the reaction is not limited to lysine and may also be tyrosine [84].

Since kinases account for less than 10% of the total captured proteins, a suitable MS data acquisition strategy is important for kinase identification [79]. With the development in instrumentation and bioinformatics, more options are available for MS data acquisition modes [85]. Using DDA mode, 100–200 kinases can be identified in enriched samples, however, kinase peptides account for only 10% of the total peptides observed [86]. The number of kinases found by DIA has increased by 21% compared to DDA [87]. PRM and MRM are target acquisition methods based on the DDA data and protein database. They are more effective in global kinome assays [88,89]. The methods have been applied in identifying a novel inhibitor target, assessing a small molecule kinase inhibitor, and mapping the kinome in response to external stimuli [90].

In combination with quantitative techniques, the chemical reactive probes have been applied to understanding regulator kinases in physiological and pathological processes. CHK1, CDK1 and CDK2 are hyperactivated in radioresistant cancer cells, which are potential effective targets for resensitizing radioresistant cancer cells [91]. An ATP affinity probe coupled with SILAC was used to assess the arsenite-induced alteration of the global kinome in human cells, and found that CDKs were key kinase responses to arsenite. The results of the Western blot were consistent with the above results [92].

### 2.4. Kinome Activity-Representing Phosphorylation Sites

The conformational changes in the functional domain of kinases can be caused by the phosphorylation of the key amino acid residues, which results in activity alteration. Thus, kinase activity can be reflected by quantifying kinome activity-representing phosphorylation sites to some extent (Figure 5) [93]. Phospho-specific antibody-based kinase assays can quantify kinase activity. Unfortunately, the throughput is too low. An antibody-based human phospho-kinase array enables a kinome high-throughput assay through a membrane-based sandwich immunoassay [94]. However, the antibody-dependent detection effect is closely related to the quality of the antibody, and the availability is limited to well-studied kinases. An alternative approach to phosphorylation sites detection is the MS-based large-scale phosphoproteomics study. Due to the low abundance of the corresponding peptides, few sites can be accurately quantified. Thus, PRM is used for quantifying the kinome activity-representing phosphorylation sites. It is sensitive and accurate in low abundance peptide analysis, and has achieved the activation profiles of 178 kinases in various biology samples [95]. Tyrosine kinases (TKs) are the main targets of kinase inhibitors; however, the coverage of the above methods on TK is limited. We find that most TKs carry at least one tyrosine residue that can be phosphorylated. The activities of TKs can be monitored by quantifying these phosphorylated tyrosine sites. Based on this, we developed a TARPL-MRM strategy to specifically characterize kinase activities, by which 85 out of 90 human TKs were covered [96]. Moreover, TARPL-MRM identified SRC as a key regulator in drug resistance, which was confirmed by Western blot. Inhibition of SRC could overcome drug resistance. Although this method can directly assess PKs activities though quantifying the phosphorylation sites, it is still in the development stage and requires optimization to improve the detection coverage of the kinome.

In summary, the above methods offer many possibilities for the determination of kinome activity. More importantly, using these methods to compare the changes in the kinome during physiological and pathological processes can reveal the underlying biological mechanisms. However, we should note that kinome results often ultimately require confirmation via biological analysis.

## 3. Potential Strategies in Kinase Spatial Assay

Kinase function is closely related to its spatial localization [97]. Alterations in the spatial localization are involved in most cellular biological processes, such as nucleocytoplasmic shuttling of p38 [98] and endocytic uptake of receptor tyrosine kinase [99]. Mislocalization of kinases is frequently associated with cellular dysfunction and diseases, including neurodegeneration, cancer and metabolic disorders [100]. Therefore, a tight control of kinase localization is an important regulatory component of cell physiology. Understanding the spatial distribution of kinases is essential for fully revealing the cell’s biology. However, traditional kinome analysis omits the spatial information in the process of tissue homogenization. The final obtained kinome information is the average level of kinase activity in the tissues or cell lysates [101]. However, it cannot reveal the differences in the spatial distribution of the kinome. In situ protein images based on FRET, confocal, or staining are effective methods for determining kinase localization, but only for individual kinases [102]. Due to their low abundance, kinome are difficult to study by traditional spatial proteomics methods. Thus, mapping the spatial distribution has been a new challenge in kinome research. Although few methods can overcome these issues, specifically, theories and methods based on MS, such as proximity labeling and low cell number proteomics, provide a potential way to overcome these limitations.

### 3.1. Proximity Labeling

Proximity labeling (PL), an advanced method for protein–protein interaction analysis, has a potential application in spatial kinome analysis. Conceptually, since the interacting proteins must be located in the same spatial location, the interaction of the kinase with the substrate can be considered as a “local” spatial proteome. They participate in various cellular processes at different spatiotemporal levels, such as in cell cycle regulation, protein synthesis and secretion, signal transduction and metabolism, and stress response [103]. Thus, a more detailed and extensive coverage of the protein kinase interactome would bring a better understanding of the biological processes and spatial localization involved in PKs. Some traditional methods for PKs’ interactome analysis, including co-immunoprecipitation (Co-IP) [104] and kinase–substrates crosslinking [105], are limited to stable complexes, or a relatively high affinity, rather than a transient or weak protein–protein interaction. PL, a novel technology for protein interactome analysis, has been developed to be directly performed in living cells under natural conditions [106]. It is conducive to the identification of hydrophobic and low-abundance protein interactions and the analysis of weak or transient protein interactions. It helps researchers better understand the complex kinase interactome and provides insights into kinase spatial information. The principle is to fuse the kinase with a tool enzyme that has a specific catalytic ligation activity. The PL enzyme is able to activate the small molecule substrate, which labels proteins with a sorting moiety (Figure 6) [107]. The labeled proteins are candidate kinase substrates, which are often difficult to be captured by conventional methods. It should be noted that complementary methods must be used to determine whether the interactors are true substrates when using PL.

The tool enzymes are critical for PL. There are many PL enzymes for protein–protein interaction identification. The characteristics of different PL enzymes are summarized and compared in Table 1. The two most commonly used are the *Escherichia coli* biotin ligase BirA mutant (BioID) and engineered ascorbate peroxidase (APEX). The BioID-based PL technique was first developed and applied to characterize the protein interactome in mammalian cells. BioID activates biotin to form biotinol-5′-AMP, which can biotinylate the lysine residues of proteins within a 10-nm radius of the target protein [108]. BioID is simple and nontoxic, but catalytic efficiency is low, typically requiring a labeling time of about 18–24 h [109]. The improved biotin ligases, such as TurboID and miniTurbo, have been adapted for PL, which is more efficient and requires less time and biotin [110]. APEX can biotinylate the tyrosine residues of proteins within a 20-nm radius when supplied with ATP, H_2_O_2_ and biotin-phenol. APEX is more efficient in labeling compared to BioID, requiring as little as 1 min to complete the biotinylated process. Unfortunately, the addition of H_2_O_2_ is toxic to cells and causes oxidative stress in the cells or tissues. Therefore, APEX2 is developed as an improved and optimized version of APEX. APEX2 has increased catalytic activity and requires less H_2_O_2_ and biotin-phenol, thus reducing the toxic effect on living cells [111]. In addition to the commonly used PL enzymes, other enzymes have been developed for PL and used in a small range, such as horseradish peroxidase (HRP), transpeptidase sortase A (SrtA) and proteasomal accessory factor A (PafA) [112].

With the development of various PL enzymes, the PL technique has broadened the study of the kinase interactome in a spatiotemporally resolved manner. It provides an important tool and pathway for the functional study of kinases, such as p38, EGFR, and PTK7 [113]. Based on PL, the interactome of p38α has been mapped under a steady or activated state, indicating the local spatial proteome difference. Novel substrates of p38 have been identified, including RNA-binding protein SRSF3 [114] and ZnF protein XPA [115], which led to novel functional insights into the signal network. Based on the high labeling efficiency of APEX, the spatiotemporal-resolved EGFR interactome has been systematically analyzed. A novel protein TFG that regulates EGFR endosomal sorting has been identified [116]. We have systematically studied the PTK7 spatial interactome and found that its mechanism in drug resistance is related to NDRG1 [117].

### 3.2. Low Cell Numbers Proteomics

The method of low cell number proteomics contributes to spatial-resolved kinases [118]. Multicellular organisms contain specialized cell types with unique functions and morphological features. The specialization is caused by differences in protein expression [119]. Different cell states may respond differently to the same exogenous signal. Even within the same cell type, the response may differ. Therefore, the spatial and cell type information can be linked with the kinome by analyzing a small number of cells. The spatial kinome can provide insight into the tissue microenvironment and uncover more precise biomarkers and new functional mechanisms [120]. Unlike single-cell analysis at the genomic and transcriptomic level, low cell number proteomic analysis cannot rely on techniques that allow amplifying trace amounts of protein [121]. Therefore, massive innovations and technologies [122] have been developed and applied in sample preparation and mass spectrometry detection, which make kinome analysis on paucicellular samples possible (Figure 7).

The sample preparation methodology must be performed as accurately as possible to reduce sample loss, especially for the low cell numbers sample. It is critical to enhance detection depth and ensure reliability and reproducibility [123]. Sample preparation first requires collecting the paucicellular sample from a variety of sources [124]. What is important is to minimize sample loss and avoid apoptosis, which may result in kinome profile alteration. For collected samples, efficient protein extraction is ideal for processing paucicellular samples with physical or biological gentle approaches [125]. Digestion efficiency has been enhanced by optimizing digestion conditions and adjusting the enzyme–protein ratio, which benefits MS detection [126]. To avoid tube-to-tube transfer and related sample losses, it is critical to perform the multistep in a single pot. Thus, multiple methods have been described to minimize sample losses, such as in-StageTip digestion (iST), filter-aided sample preparation (FASP), and single-pot solid-phase-enhanced sample preparation (SP3) [127,128]. These methods have been used for the preparation of paucicellular samples, even a single cell.

High-performance peptide separation and sensitive data acquisition have been used to improve kinome coverage significantly in MS detection [129]. The use of narrow columns for peptide separation can reduce the peak width and increase the concentration of the eluted peptide. It contributes to improve the ionic strength and increase the sensitivity [130]. The stationary phase, which is derived with a weakly basic ionizable silane or amine-bridged organosilane precursor, has been used to improve the peak shape effectively [130]. Targeted data acquisition via SureQuant, Pseudo-PRM, TOMAHAQ, and ScoutMRM enables greatly enhanced sensitivity [131]. BOOST, as an alternative method, is based on TMT labeling and takes advantage of signal enhancement in the MS1 level. It enhances the identification and quantification of low-abundance peptides by using a “carrier channel”, which contains a relatively high sample volume [132]. This method has been used for phosphoproteome study, increasing the sensitivity of the assay by 20-fold or so [133]. The kinome activity assay method can be combined with low cell number proteomics easily, and it can be applied in spatial-resolved kinome analysis.

## 4. Conclusions

A variety of reliable methodologies for profiling kinome activity have been developed on the basis of monitoring the phosphorylation of substrates or kinases with high-throughput. The application of these approaches has provided much insight into the understanding of biological and pathological processes, which has led to the revelation of the master kinase in disease development. These approaches can improve individual patient medicine and minimize the risk of adverse effects. In addition, spatial kinome analysis is an important method to elucidate the biological function of kinase. The current technologies used in proximity labeling and low cell number proteomics have the potential to be applied in spatial kinome analysis. Spatial kinome analysis is complementary to conventional kinome analysis, rather than merely improving sensitivity. The spatial kinome analysis expounds the abnormal kinase and reveals its molecular mechanism more systematically and comprehensively, which is helpful for druggable target discovery. Conventional kinome analysis provides an important guiding tool for the design, screening and evaluation of kinase inhibitors. In summary, they can facilitate the development of kinase inhibitors and promote the development of precision medicine.

## Figures and Tables

**Figure 1 cells-12-00925-f001:**
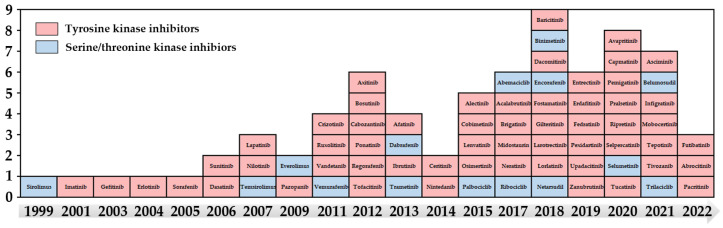
US Food and Drug Administration (FDA)-approved small molecular inhibitors that target kinases.

**Figure 2 cells-12-00925-f002:**
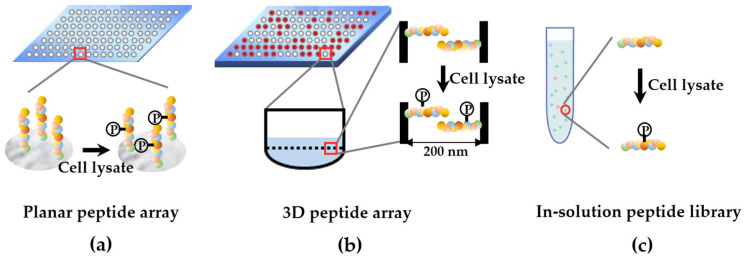
Overview of kinome profiling approaches based on the substrate peptide library.

**Figure 3 cells-12-00925-f003:**
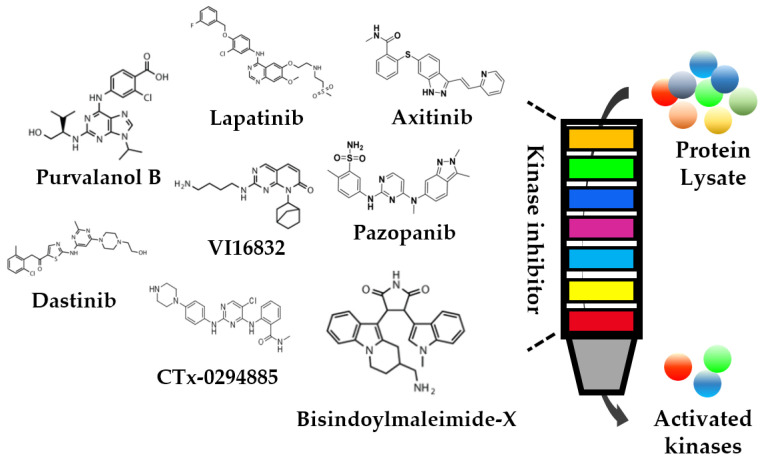
Workflow of kinome enrichment. Protein lysate is passed through an inhibitor conjugated beads column. The activated kinases are enriched and eluted in the column.

**Figure 4 cells-12-00925-f004:**
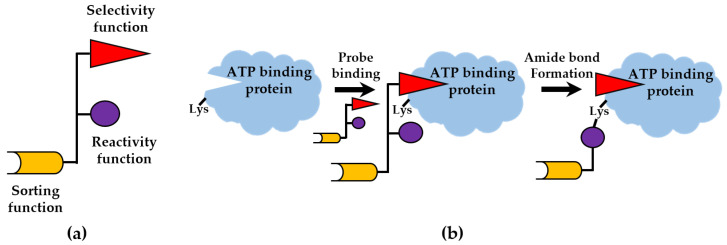
The principle of a chemical reactive probe. (**a**) Depiction of schematic kinome capture compounds; (**b**) A schematic diagram showing the conjugation between reactive ATP affinity probes with an ATP-binding protein including kinase.

**Figure 5 cells-12-00925-f005:**
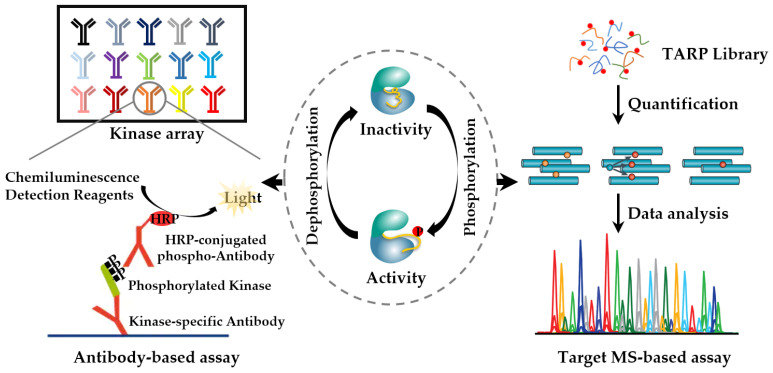
Quantification of kinase activity-representing phosphorylation sites to determine kinome activity. Antibody-based kinase array and target MS-based methods are used for the quantification of phosphorylated kinase.

**Figure 6 cells-12-00925-f006:**
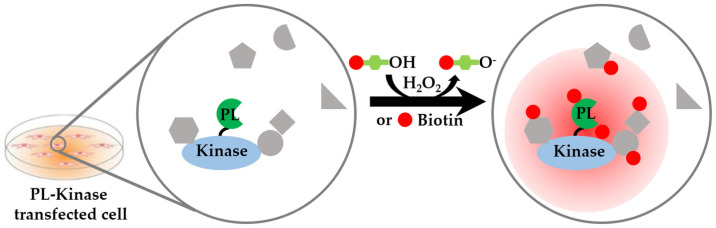
Kinase–substrate interaction analysis based on proximity labeling. PL enzymes are fused to the kinase of interest, which is expressed in the cells. The PL enzymes catalyze biotin-phenol or biotin into reactive biotin, which diffuses and labels proximal proteins.

**Figure 7 cells-12-00925-f007:**
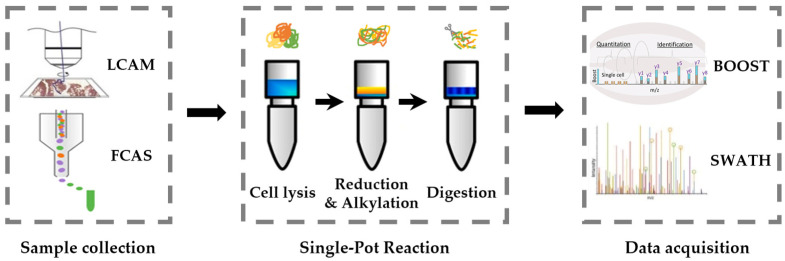
Schematic workflow of the low cell number proteome method. Samples are collected from tissue or cells via LCAM or FACS, then treated in a single pot with three main steps—cell lysis, reduction and alkylation, and digestion. Peptides are detected by the LC-MS/MS with a sensitivity data acquisition mode, such as BOOST and SWATH.

**Table 1 cells-12-00925-t001:** Overview of PL enzymes.

Enzyme	Size (kDa)	Labeling Time	Labeling Radius (nm)	Modification Sites	Substrate	Advantages	Limitations
APEX	28	1 min	~20	Tyr, Trp, Cys, His	Biotin-phenol H_2_O_2_	High temporal resolution; versatility for both protein and RNA labeling	Limited application in vivo because of the toxicity of H_2_O_2_
APEX2	28	1 min	~20	Tyr, Trp, Cys, His	Biotin-phenol+H_2_O_2_		
BioID	35	18 h	~10	Lys	Biotin	Non-toxic for in vivo applications	Poor temporal resolution because of the low activity
BioID2	27	18 h	~10	Lys	Biotin		
TurboID	35	10 min	~10	Lys	Biotin	Highest activity biotin ligase; Non-toxic for in vivo applications	Potentially less control of the labeling window because of high biotin affinity
miniTurbo	28	10 min	~10	Lys	Biotin		Lower activity and stability as compared to TurboID.

## Data Availability

Not applicable.

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
