# Peer review of "Mapping the Protein Kinome: Current Strategy and Future Direction"

_cells, 2023, doi:10.3390/cells12060925_

Round 1

Reviewer 1 Report

The manuscript presents a review on the widely used methods for kinome analysis and the challenges encountered in their applications. The paper reviewed a variety of reliable methodology for profiling kinome activity that have been developed on the base of monitoring the phosphorylation of substrates or kinases with high-through-put, and concluded the applications of these approaches have provided insight into the understanding of biological and pathological processes, which led to reveal the master kinase in disease development. The MS adequately described up to date findings and conclusion consistent with the evidence from published studies. The references are appropriate and relevant to the research.

Author Response

On behalf of my co-authors, I would like to express our sincere appreciations of your review work on our manuscript.

Reviewer 2 Report

The authors of this article reviewed the technologies used in conventional and spatial kinome analysis and commented on the advantages and disadvantages of these technologies in their applications. The authors have clearly described the principles of each method and the importance of kinome analysis in biological settings.  It will improve the article significantly if the application of these technologies in physiological and pathological processes has been commented on. It is also interesting to compare the outcomes of using these techniques with the results of using biological assays, e.g. when determining the localization of a protein in a cell, how well the spatial kinome analysis will do compared with the in situ protein staining? How will a kinase activity be determined through kinome analysis compared to a direct kinase assay? 

Reviewer 3 Report

This manuscript by Zhanwu Hou and Huadong Liu provides a comprehensive review on the strategies and techniques which allow us to identify and analyze kinomes in different cells and tissues. Many techniques such as using kinase substrate peptides or conjugated small molecular inhibitors, among other approaches for kinome identification and characterization have been reviewed. The content of the paper is suitable for publication in CELLS. Readers would definitely benefit from reading this review paper to learn more about the current strategies to map kinome. 

However, this manuscript contains too many English grammar mistakes and other writing errors. I would suggest the authors to improve the writing of the paper before publication in CELLS. The following lists some of the grammar issues that I can find which need revision:

Minor revisions: 

1. Line 48, of academia and industry?

2. Line 79-80, initial rate (Vo)? Michaelis-Mention constants (Km, Kcat, Vm)?

3. Line 92, categories:

4. Line 111, the peptide library is incubated with

5. Line 113-114, radioautography, western blotting with phosphorylation-specific antibodies, and fluorescence spectroscopy?

6. Line 123-123, generation

7. Line 139, As the data show similarity

8. Line 168, have been identified?

9. Line 171, highly selective

10. Line 177, 60 cell lines using MIBs which is then verified by MS-based protein identification.

11. Line 179, kinases

12. Line 187, which is benefited from a wide … ?

13. Line 216, probes

14. Line 223, ATP binding domain?

15. Line 224, do not understand this: “which can specifically and broadly recognize ATP” specifically binds to ATP but not other nucleotides?

16. Line 245, ATP-competitive kinase inhibitor? Or ATP analog?

17. Line 150, ATP binding domain?

18. Line 268, conformational changes in functional domain

19. Line 271, antibody-based

20. Line 272, can quantify kinase activity?

21. Line 280, low abundance peptide analysis?

22. Line 281, …kinase inhibitors; however, 

23, Line 283, can be monitored by … ?

24, Line 285, to specifically characterize kinase activities, by which 85 out of 90 human …

25. Line 305, few methods can overcome/circumvent ?

26. Line 319, protein-protein interaction?

27. Line 320, has been developed to be directly performed in 

28. Line 333, the kinase of interest is expressed…

29. Line 337, the two most commonly 

30. Line 348, H2O2 is toxic to cells …

31. Line 349, APEX2 is developed as an improved and optimized version of APEX?

32. Line 356, spatiotemporally resolved manner?

33. Line 358, under steady or activated state

34. Line 365, is related to NDRG1

35. Line 392, Digestion efficiency has been…

36. Line 394, (delete Whatever,) To avoid tube-to-tube…

37. Line 404, The stationary phase which is derived from week basic ionizable silane …

38. Line 405, has been used…

39. Line 415, methodologies 

40. Line 416, on the basis of 

41, Line 418, ..have led to the revelation of the master kinases in…

42. Line 419, do not understand this sentence.

43, Line 420, requires less loss in sample …

44. Line 422, and less tissue volume.
